**Brief Communication**

# Deep-learning-based gene perturbation effect prediction does not yet outperform simple linear baselines

**Constantin Ahlmann-Eltze** [1,2,3] ✉, **Wolfgang Huber** [2] **& Simon Anders** [1]

Recent research in deep-learning-based foundation models promises to learn representations of single-cell data that enable prediction of the effects of genetic perturbations. Here we compared five foundation models and two other deep learning models against deliberately simple baselines for predicting transcriptome changes after single or double perturbations. None outperformed the baselines, which highlights the importance of critical benchmarking in directing and evaluating method development.

The success of large language models in knowledge representation has spawned efforts to apply the foundation model concept to biology[1–3]. Several single-cell foundation models trained on transcriptomics data from millions of single cells have been published[4–6]. Two recent models—scGPT[7] and scFoundation[8]—claim to be able to predict gene expression changes caused by genetic perturbations.

In the present study, we benchmarked the performance of these models against GEARS[9] and CPA[10] and against deliberately simplistic baselines. To provide additional perspective, we also included three single-cell foundation models—scBERT[4], Geneformer[5] and UCE[6]—that were not explicitly designed for this task but can be repurposed for it by combining them with a linear decoder that maps the cell embedding to the gene expression space. In the figures, we marked their results with an asterisk.

We first assessed prediction of expression changes after double perturbations. We used data by Norman et al.[11], in which 100 individual genes and 124 pairs of genes were upregulated in K562 cells with a CRISPR activation system (Extended Data Fig. 1). The phenotypes for these 224 perturbations, plus the no-perturbation control, are logarithm-transformed RNA sequencing expression values for 19,264 genes.

We fine-tuned the models on all 100 single perturbations and on 62 of the double perturbations and assessed the prediction error on the remaining 62 double perturbations. For robustness, we ran each analysis five times using different random partitions.

For comparison, we included two simple baselines: (1) the 'no change' model that always predicts the same expression as in the control condition and (2) the 'additive' model that, for each double perturbation, predicts the sum of the individual logarithmic fold changes (LFCs). Neither uses the double perturbation data.

All models had a prediction error substantially higher than the additive baseline (Fig. 1a,b). Here, prediction error is the $L_2$ distance between predicted and observed expression values for the 1,000 most highly expressed genes. We also examined other summary statistics, such as the Pearson delta measure, and $L_2$ distances for other gene subsets: the $n$ most highly expressed or the $n$ most differentially expressed genes, for various $n$. We got the same overall result (Extended Data Fig. 2).

Next, we considered the ability of the models to predict genetic interactions. Conceptually, a genetic interaction exists if the phenotype of two (or more) simultaneous perturbations is 'surprising'. We operationalized this as double perturbation phenotypes that differed from the additive expectation more than expected under a null model with a Normal distribution (Extended Data Fig. 3 and Methods). Using the full dataset, we identified 5,035 genetic interactions (out of potentially 124,000) at a false discovery rate of 5%.

We then obtained genetic interaction predictions from each model by computing, for each of its 310,000 predictions (1,000 read-out genes and 62 held-out double perturbations across five test–training splits), the difference between predicted expression and additive expectation, and, if that difference exceeded a given threshold $D$, we called a predicted interaction. We then computed, for all possible choices of $D$, the true-positive rate (TPR) and the false discovery proportion, which resulted in the curves shown in Fig. 1c. The additive model did not compete as, by definition, it does not predict interactions.

None of the models was better than the 'no change' baseline. The same ranking of models was observed when using other metrics (Extended Data Fig. 4).

To further dissect this finding, we classified the interactions as 'buffering', 'synergistic' or 'opposite' (Fig. 1d,e and Methods). All models

[1]BioQuant, University of Heidelberg, Heidelberg, Germany. [2]Genome Biology Unit, European Molecular Biology Laboratory (EMBL), Heidelberg, Germany. [3]Present address: UCL Cancer Institute, London, UK. ✉e-mail: constantin.ahlmann@embl.de

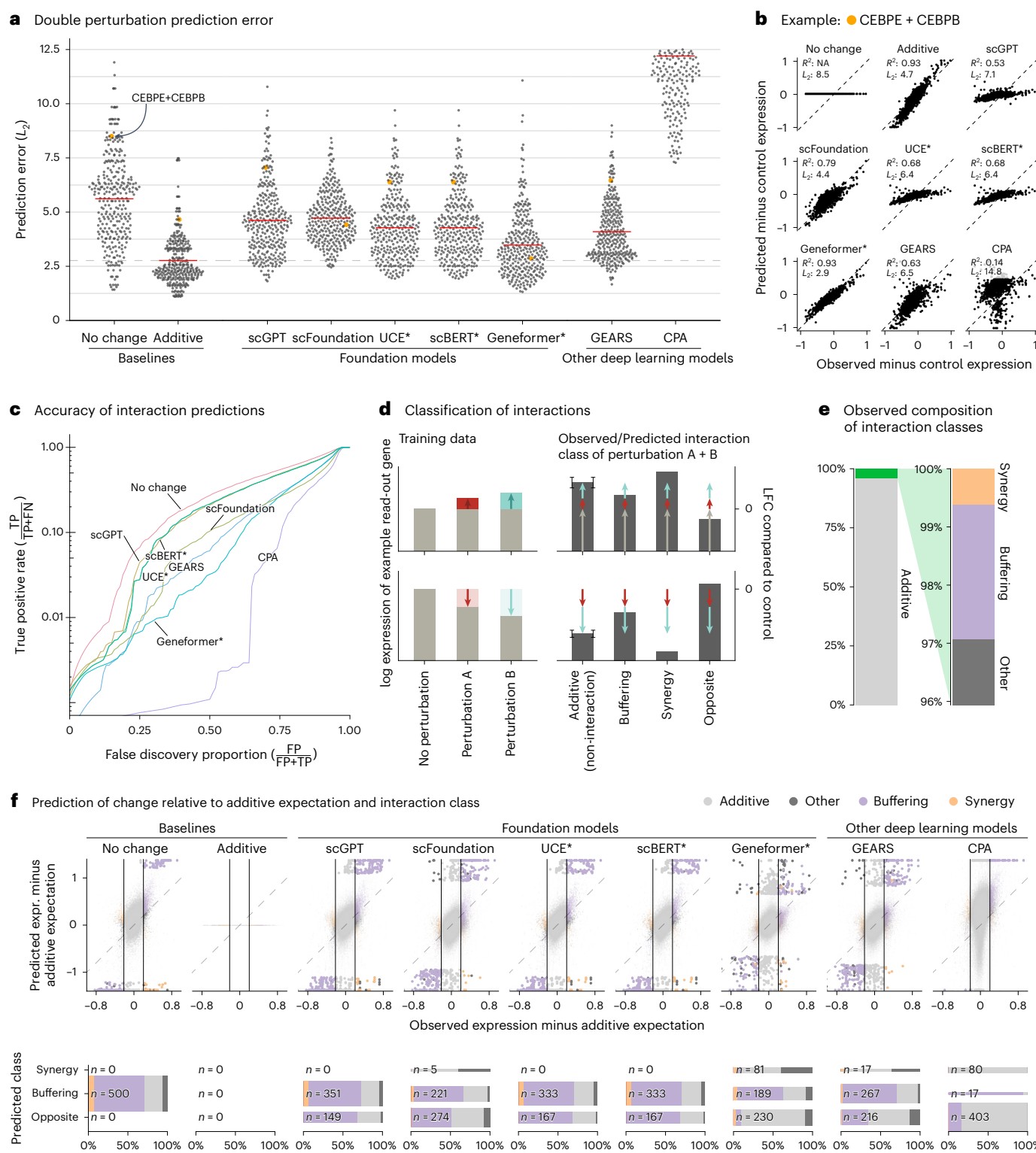

**Fig. 1 | Double perturbation prediction. a**, Beeswarm plot of the prediction errors for 62 double perturbations across five test–training splits. The prediction error is measured by the $L_2$ distance between the predicted and the observed expression profile of the $n = 1{,}000$ most highly expressed genes. The horizontal red lines show the mean per model, which, for the best-performing model, is extended by the dashed line. **b**, Scatterplots of observed versus predicted expression from one example of the 62 double perturbations. The numbers indicate error measured by the $L_2$ distance and the Pearson delta ($R^2$). **c**, TPR (recall) of the interaction predictions as a function of the false discovery proportion. FN, false negative; FP, false positive; TP, true positive. **d**, Schematic of the classification of interactions based on the difference from the additive expectation (the error bars show the additive range). **e**, Bar chart of the composition of the observed interaction classes. **f**, Top: scatterplot of observed versus predicted expression compared to the additive expectation. Each point is one of the 1,000 read-out genes under one of the 62 double perturbations across five test–training splits. The 500 predictions that deviated most from the additive expectation are depicted with bigger and more saturated points. Bottom: mosaic plots that compare the composition of highlighted predictions from the top panel stratified by the interaction class of the prediction. The width of the bars is scaled to match the number of instances. Source data for Fig. 1 are provided. expr., expression.

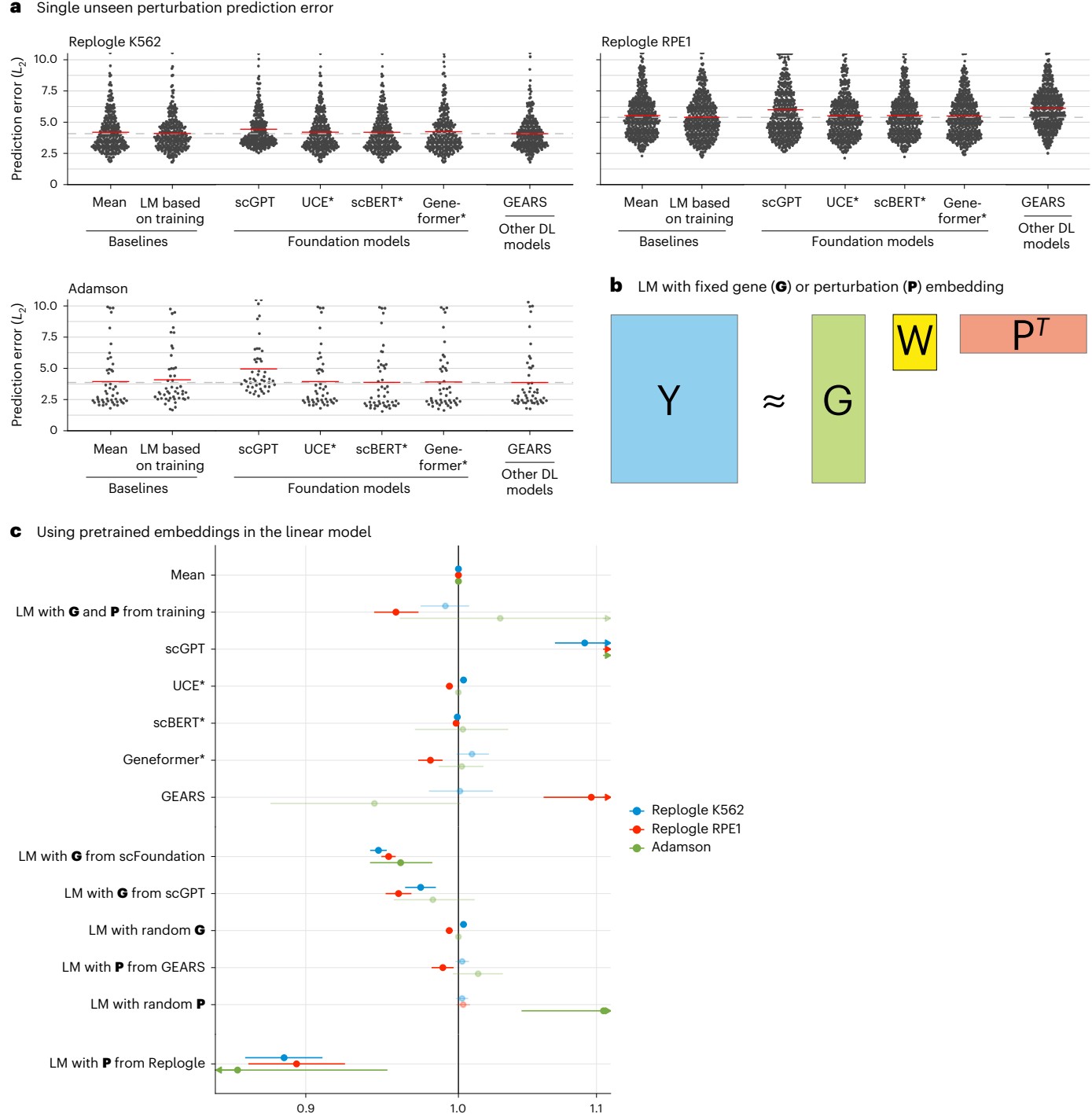

**Fig. 2 | Single perturbation prediction. a**, Beeswarm plot of the prediction errors for 134, 210 and 24 unseen single perturbations across two test–training splits (Methods). The prediction error is measured by the $L_2$ distance between the mean predicted and observed expression profile of the $n = 1{,}000$ most highly expressed genes. The horizontal red lines show the mean per model, which, for the best-performing model, is extended by the dashed line. DL, deep learning; LM, linear model. **b**, Schematic of the LM and how it can accommodate available gene (**G**) or perturbation (**P**) embeddings. **c**, Forest plot comparing the performance of all models relative to the error of the 'mean' baseline. The point ranges show the overall mean and 95% confidence interval of the bootstrapped mean ratio between each model and the baseline for 134, 210 and 24 unseen single perturbations across two test–training splits. The opacity of the point range is reduced if the confidence interval contains 0. Source data for Fig. 2 are provided.

mostly predicted buffering interactions. The 'no change' baseline cannot, by definition, find synergistic interactions, but also the deep learning models rarely predicted synergistic interactions, and it was even rarer that those predictions were correct (Fig. 1f).

To our surprise, we often found the same pair of hemoglobin genes (*HBG2* and *HBZ*) among the top predicted interactions, across models and double perturbations (Extended Data Fig. 5). Examining the data, we noted that all models except Geneformer and scFoundation

predicted LFC ≈ 0—like the 'no change' baseline—for the double perturbation of these two genes, despite their strong individual effects (Extended Data Fig. 6). More generally, we noted that, for most genes, the predictions of scGPT, UCE and scBERT did not vary across perturbations, and those of GEARS and scFoundation varied considerably less than the ground truth (Extended Data Fig. 7).

GEARS, scGPT and scFoundation also claim the ability to predict the effect of unseen perturbations. GEARS uses shared Gene Ontology[12] annotations to extrapolate from the training data, whereas the foundation models are supposed to have learned the relationships between genes during pretraining to predict unseen perturbations.

To benchmark this functionality, we used two CRISPR interference datasets by Replogle et al.[13] obtained with K562 and RPE1 cells and a dataset by Adamson et al.[14] obtained with K562 cells (Extended Data Fig. 1).

As a baseline, we devised a simple linear model. It represents each read-out gene with a $K$-dimensional vector and each perturbation with an $L$-dimensional vector. These vectors are collected in the matrices $\mathbf{G}$, with one row per read-out gene, and $\mathbf{P}$, with one row per perturbation. $\mathbf{G}$ and $\mathbf{P}$ are either obtained as dimension-reducing embeddings of the training data (Methods) or provided by an external source (see below). Then, given a data matrix $\mathbf{Y}_{\text{train}}$ of gene expression values, with one row per read-out gene and one column per perturbation (that is, per condition pseudobulk of the single-cell data), the $K \times L$ matrix $\mathbf{W}$ is found as

$$\underset{\mathbf{W}}{\arg\min} \, ||\mathbf{Y}_{\text{train}} - (\mathbf{GWP}^T + \boldsymbol{b})||_2^2, \qquad (1)$$

where $\boldsymbol{b}$ is the vector of row means of $\mathbf{Y}_{\text{train}}$ (Fig. 2b).

We also included an even simpler baseline, $\boldsymbol{b}$, the mean across the perturbations in the training set, following the preprints by Kernfeld et al.[15] and Csendes et al.[16] that appeared while this paper was in revision.

None of the deep learning models was able to consistently outperform the mean prediction or the linear model (Fig. 2a and Extended Data Fig. 8). We did not include scFoundation in this benchmark, as it required each dataset to exactly match the genes from its own pretraining data, and, for the Adamson and Replogle data, most of the required genes were missing. We also did not include CPA, as it is not designed to predict the effects of unseen perturbations.

Next, we asked whether we could find utility in the data representations that GEARS, scGPT and scFoundation had learned during their pretraining. We extracted a gene embedding matrix $\mathbf{G}$ from scFoundation and scGPT, respectively, and a perturbation embedding matrix $\mathbf{P}$ from GEARS. The above linear model, equipped with these embeddings, performed as well or better than scGPT and GEARS with their in-built decoders (Fig. 2c). Furthermore, the linear models with the gene embeddings from scFoundation and scGPT outperformed the 'mean' baseline, but they did not consistently outperform the linear model using $\mathbf{G}$ and $\mathbf{P}$ from the training data.

The approach that did consistently outperform all other models was a linear model with $\mathbf{P}$ pretrained on the Replogle data (using the K562 cell line data as pretraining for the Adamson and RPE1 data and the RPE1 cell line for the K562 data). The predictions were more accurate for genes that were more similar between K562 and RPE1 (Extended Data Fig. 9). Together, these results suggest that pretraining on the single-cell atlas data provided only a small benefit over random embeddings, but pretraining on perturbation data increased predictive performance.

In summary, we presented prediction tasks where current foundation models did not perform better than deliberately simplistic linear prediction models, despite significant computational expenses for fine-tuning the deep learning models (Extended Data Fig. 10). As our deliberately simple baselines are incapable of representing realistic biological complexity, yet were not outperformed by the foundation models, we conclude that the latter's goal of providing a generalizable representation of cellular states and predicting the outcome of not-yet-performed experiments is still elusive.

The publications that presented GEARS, scGPT and scFoundation included comparisons against GEARS and CPA and against a linear model. Some of these comparisons may have happened to be particularly 'easy'. For instance, CPA was never designed to predict effects of unseen perturbations and was particularly uncompetitive in the double perturbation benchmark. The linear model used in scGPT's benchmark appears to have been set up such that it reverts to predicting no change over the control condition for any unseen perturbation.

Our results are in line with previously published benchmarks that assessed the performance of foundation models for other tasks and found negligible benefits compared to simpler approaches[17–19]. Our results also concur with two previous studies showing that simple baselines outperform GEARS for predicting unseen single or double perturbations[20,21]. Since the release of our paper as a preprint, several other benchmarks[15,16,22–27] were released that also show that deep learning models struggle to outperform simple baselines. Two of these preprints[15,16] suggested an even simpler baseline than our linear model (equation (1)), namely, to always predict the overall average, and we have included this idea here.

One limitation of our benchmark is that we used only four datasets. We chose these as they were used in the publications presenting GEARS, scGPT and scFoundation. Another limitation is that all datasets are from cancer cell lines, which, for example, Theodoris et al.[5] excluded from their training data because of concerns about their high mutational burden. We also did not attempt to improve the original quality control, for example, by excluding perturbations that did not affect the expression of their own target gene and, thus, might not have worked as intended.

Deep learning is effective in many areas of single-cell omics[28,29]. However, prediction of perturbation effects still remains an open challenge, as our present work shows. We expect that increased focus on performance metrics and benchmarking will be instrumental to facilitate eventual success in applying transfer learning to perturbation data.

## Online content

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

## Methods

### Data

We ran the double perturbation benchmark on the data produced by Norman et al.[11] and reprocessed by scFoundation. For the single gene perturbation benchmarks, we used the data from Adamson et al.[14] and Replogle et al.[13] as provided by GEARS (details in 'Data availability').

### Software versions and parameters

We ran GEARS version 0.1.2, scGPT version 0.2.1, scFoundation (which is built on top of a GEARS version 0.0.2 fork), CPA version 0.8.8, Geneformer version 0.1.0, scBERT from commit hash 262fd4b9 with model weights provided by the authors and UCE at commit hash 8227a65c. We used each model, as much as possible, with their default parameters. All scripts that were used to predict the expression changes are available on GitHub (https://github.com/const-ae/linear_perturbation_prediction-Paper/tree/main/benchmark/src).

- GEARS and scFoundation provide a straightforward application programming interface (API) to predict the expression change after perturbation. We limited the fine-tuning time to 3 days, which meant that we trained scFoundation for five epochs.
- For scGPT, we used the same parameters and code as in their tutorial for perturbation prediction.
- For CPA, we used the code from their tutorial on how to predict combinatorial CRISPR perturbations on the Norman dataset.
- For Geneformer, we fine-tuned the provided model by predicting the perturbation labels of the training data. We then used the built-in in silico perturbation functionality to calculate the perturbed embedding.
- UCE is designed for zero-shot use, which means that it does not need to be fine-tuned. We report results from the four-layer version of UCE (as we found no performance difference between the four-layer and 33-layer versions). UCE does not provide functionality for in silico perturbation, so we calculated the post-perturbation embedding by taking the expression matrix for the unperturbed cells and overwrote the rows for the genes that we wanted to perturb with the values from the ground truth expression matrix. We, thus, tried to ensure that we tested the model under the best conditions, accepting that test data leakage could theoretically give the model an advantage over the other models.
- We fine-tuned scBERT on predicting the perturbation labels of the training data. We then used the same approach to calculate the embedding after in silico perturbation that we used for UCE.

To predict the expression changes from the embeddings of Geneformer, UCE and scBERT, we added a linear decoder to the models. We fitted a ridge regression model that predicted the gene expression of the perturbed cells from the perturbed embeddings of the training data. We then used that ridge regression to predict the gene expression of the test data from the corresponding perturbed embeddings and continued with the mean of the predicted values per perturbation.

To reduce the probability that we understate the performance for any of the models, due to wrong or suboptimal operation by ourselves, we reached out to the original authors of the benchmarked models and asked them to review our code. The authors of CPA perceived a problem with our code and submitted a fix; however, as the new code had worse performance than the original version, here we report results of the original code.

### Double perturbation benchmark setup

For the double perturbation benchmark, we split the data into test and training sets. We assigned all single-gene perturbations and a randomly chosen half of the double perturbations to the training set and used the other half of the double perturbations as the test set. To reduce stochastic effects on our results, we repeated the whole procedure, including the random test–training splitting, five times.

We used two baseline models: 'no change' and 'additive'. The 'no change' model 'predicted', for each double perturbation, the expression values seen in the control condition ($y^\varnothing$). The 'additive' model predicts the expression after a double perturbation of genes A and B as

$$\hat{y}^{\text{add}} = y^A + y^B - y^\varnothing, \tag{2}$$

where $y^A$ and $y^B$ are the mean observed expression vectors for the single perturbation of genes A and B, respectively.

We defined genetic interactions as follows. For each of the 124 double perturbations and the 1,000 read-out genes, we computed the difference between the observed expression value and the additive expectation. These values showed a mixture distribution composed of a large component with a single narrow peak around 0 (corresponding to a majority of non-interactions) and a smaller component consisting of two pronounced tails on either side (corresponding to interactions) (Extended Data Fig. 3). To decompose this mixture, we used Efron's empirical null approach[30], as implemented in the 'locfdr' package (version 1.1-8).

We further classified the interactions, if the two individual LFCs had the same sign, as:

- 'buffering', if the LFC was between 0 and the additive expectation
- 'synergistic', if it exceeded the additive expectation
- 'opposite', if its sign differed from that of the individual perturbations

If the individual effects were in opposite directions, 'other'. According to this classification, 2.3% of the read-out gene expression values across all double perturbation were buffering interactions; 0.6% were synergistic; and zero were in the opposite direction of the individual perturbations.

### Single perturbation benchmark setup

For the single perturbation benchmark, we used the data as provided by GEARS and also used its 'simulation' test–training splitting procedure, which we repeated twice.

To predict the effects of unseen single perturbations, we used two baselines. The 'mean' model calculated the mean of the expression values in the training data. The 'linear model' is implied in equation (1). We set $b$ to the row means of the training data ($b = 1/N\sum_i Y^{\text{train}}_{:i}$). We find $G$ and $P$ as follows. Perform a principal component analysis (PCA) on $Y_{\text{train}}$ and use the top $K$ principal components for $G$. Then, subset this $G$ to only the rows corresponding to genes that were perturbed in the training data (and, hence, appear as columns in $Y^{\text{train}}$) and use the resulting matrix for $P$.

Then, we find $W$ using the normal equations

$$W = (G^T G + \lambda I)^{-1} G^T (Y_{\text{train}} - b) P (P^T P + \lambda I)^{-1}, \tag{3}$$

where we use a ridge penalty of $\lambda = 0.1$ for numerical stability. Having found a $W$, we can use it for prediction, $\hat{Y} = GW\bar{P}^T + b$, where now $\bar{P}$ is the matrix formed by the rows of $G$ corresponding to genes perturbed in the test data.

For the single perturbation analysis, not all models were able to predict the expression change for all unseen perturbations. For example, the linear model with $G$ and $\bar{P}$ from the training data could only predict perturbations where the target genes were also part of the read-out genes. To evaluate all models on a consistent set of perturbations, we restricted our analysis to those perturbations for which we had predictions from all models (73 perturbations for Adamson, 398 for Replogle K562 and 629 for Replogle RPE1).

We converted GEARS' Gene Ontology annotations into a perturbation embedding $P$ by computing a spectral embedding[31,32] of the pathway membership matrix. We extracted the gene embedding $G$ from scGPT following their tutorial on gene regulatory inference. For scFoundation, we extract $G$ directly from the pretrained model weights ('pos_emb.weight'). For the

linear model with **P** from the Replogle data, we fitted a 10-dimensional PCA on the columns of the matrix with the perturbation means of the reference data. We fitted all linear models as described in the main text with $K = 10$; if **G** or **P** was provided, we simply replaced the estimate from the training data with the provided matrix before calculating **W**.

The additive model is a special case of the linear model (equation (1)) where the gene embedding is simply the single perturbation data, without any further transformation or dimension reduction ($\mathbf{G} = \mathbf{Y}^{\text{single}}$); the perturbation embedding **P** is a binary coding, where each column vector has 1s in the rows of the perturbed genes and is 0 otherwise; and $W$ is an identity matrix and $\boldsymbol{b} = -\boldsymbol{y}^{\varnothing}$.

### Evaluation metrics

We measured the prediction error using the distance $L_2(\hat{\boldsymbol{y}}, \boldsymbol{y}) = \sqrt{\sum_g (\hat{y}_g - y_g)^2}$ (also called root mean squared error) between the observed expression values and predictions for the 1,000 most highly expressed genes in the control condition. We also calculated the Pearson delta correlation metric, as suggested by Cui et al.[7]: PearsonDelta $(\hat{\boldsymbol{y}}, \boldsymbol{y}) = \text{cor}(\hat{\boldsymbol{y}} - \boldsymbol{y}^{\varnothing}, \boldsymbol{y} - \boldsymbol{y}^{\varnothing})$. Unlike the $L_2$ distance, the Pearson delta metric does not penalize predictions that are consistently too small or too large in amplitude and, thus, prioritizes correct prediction of the direction of the expression change.

For the double perturbation data, we assess the TPR (recall) as a function of the false discovery rate. First, we find the order statistic of absolute difference of predictions and additive expectation across all test perturbations ($\boldsymbol{j} = \text{argsort}(\text{abs}(\hat{\mathbf{Y}} - \hat{\mathbf{Y}}^{\text{add}}))$), where $\hat{\mathbf{Y}}$ is the matrix of the predictions for all genes and perturbations and $\hat{\mathbf{Y}}^{\text{add}}$ are the additive expectations.

The false discovery proportion (FDP) at position $l \in \{1, \cdots, N\}$ for a threshold $u$, which separates the interactions from the non-interactions, is

$$\text{FDP}_l = \frac{\sum_{i=1}^{l} \mathbf{1}(\text{abs}(\mathbf{Y} - \hat{\mathbf{Y}}^{\text{add}})_{j_i} < u)}{l} \quad (4)$$

and the TPR is

$$\text{TPR}_l = \frac{\sum_{i=1}^{l} \mathbf{1}(\text{abs}(\mathbf{Y} - \hat{\mathbf{Y}}^{\text{add}})_{j_i} \geq u)}{\sum_{i=1}^{N} \mathbf{1}(\text{abs}(\mathbf{Y} - \hat{\mathbf{Y}}^{\text{add}})_i \geq u)}, \quad (5)$$

where **Y** is the matrix of observed value and $N$ is the product of the number of genes and perturbations. The indicator function $\mathbf{1}(\cdot)$ counts how often the observed values **Y** deviate enough from the additive expectation so that the observations are considered an interaction. The order statistic $\boldsymbol{j}$ ensures that we consider the gene–perturbation pairs first, where the model prediction deviates most from the additive expectation.

Lastly, we find the order statistic of the FDPs ($\boldsymbol{s} = \text{argsort}(\text{FDP})$) and plot the tuples $1, \cdots, N$

$$(\text{FDP}_{s_l}, \max_i^{1 \cdots l}(\text{TPR}_{s_i})). \quad (6)$$

An advantage of considering here the false discovery versus true-positive curve, compared with the precision-recall or the receiver operator curve, is that it provides a direct assessment of which fraction of interactions a model identifies for a fixed fraction of false positives.

### Reporting summary

Further information on research design is available in the Nature Portfolio Reporting Summary linked to this article.

### Data availability

All datasets used in this paper are publicly available: the Norman et al.[11] was downloaded via scFoundation (https://figshare.com/ndownloader/files/44477939); the Adamson et al.[14] was downloaded via GEARS (https://dataverse.harvard.edu/api/access/datafile/6154417); the Replogle et al.[13] K562 was downloaded via GEARS (https://dataverse.harvard.edu/api/access/datafile/7458695); and the Replogle et al.[13] RPE1 was also downloaded via GEARS (https://dataverse.harvard.edu/api/access/datafile/7458694). Source data for Figs. 1 and 2 and Extended Data Figs. 1–3, 5, 6 and 8–10 are provided.

### Code availability

The code to reproduce the analyses presented here and details about the software package versions are available at github.com/const-ae/linear_perturbation_prediction-Paper, which we also archived on Zenodo[33]. The Zenodo repository also contains the results of the intermediate calculations needed to reproduce all figures.

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

### Acknowledgements

We thank an anonymous reviewer of one of our previous papers[34] for the suggestion to compare foundation models against simple linear models, which eventually led to this work. This work has been supported by the European Research Council (Synergy Grant DECODE under grant agreement number 810296) and by the Klaus Tschira Foundation (grant number 00.022.2019).

### Author contributions

C.A.E., W.H. and S.A. conceived the study and wrote the paper. C.A.E. performed the computations, with feedback from W.H. and S.A.

### Funding

### Competing interests

The authors declare no competing interests.

### Additional information

**Extended data** is available for this paper at https://doi.org/10.1038/s41592-025-02772-6.

**Correspondence and requests for materials** should be addressed to Constantin Ahlmann-Eltze.

**(A) Table: Dataset overview**

| Dataset (cell line) | Size (Genes × Cells) | Perturbations |
|---|---|---|
| Norman (K562) | 19 264 × 84 143 | 124 double and 100 single perturbations |
| Replogle (K562) | 5 000 × 162 264 | 1 087 single perturbations |
| Replogle (RPE1) | 5 000 × 161 423 | 1 534 single perturbations |
| Adamson (K562) | 5 060 × 65 899 | 81 single perturbations |

**(B) UMAP of the datasets**

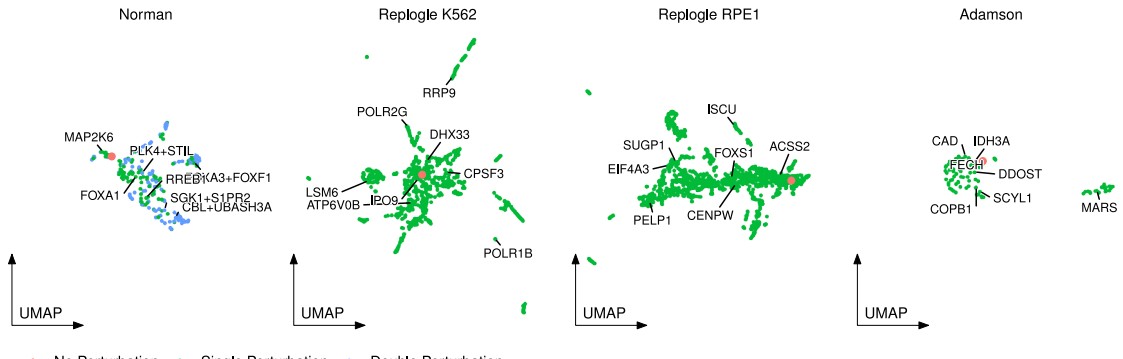

**(C) Change of perturbation target gene expression**

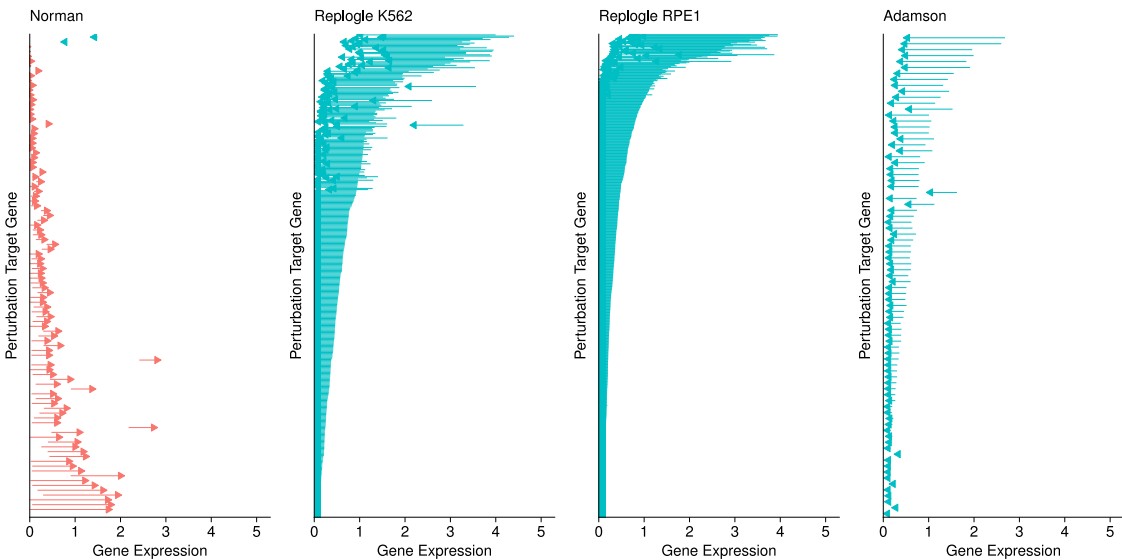

**Extended Data Fig. 1 | Dataset Overview. (a)** Table with the size of the data and the number of perturbations. **(b)** UMAP on the perturbations per dataset (aggregated to the mean per perturbation). The position of the control condition without perturbation is shown in red, and a random selection of perturbations is labeled. **(c)** Change in the expression of the target gene of each perturbation. The base of the arrow indicates the expression without perturbation, and the tip indicates the expression after perturbation. For genes targeted multiple times in the Norman dataset, we show the average expression after perturbation.

**(A) Double perturbation prediction correlation**

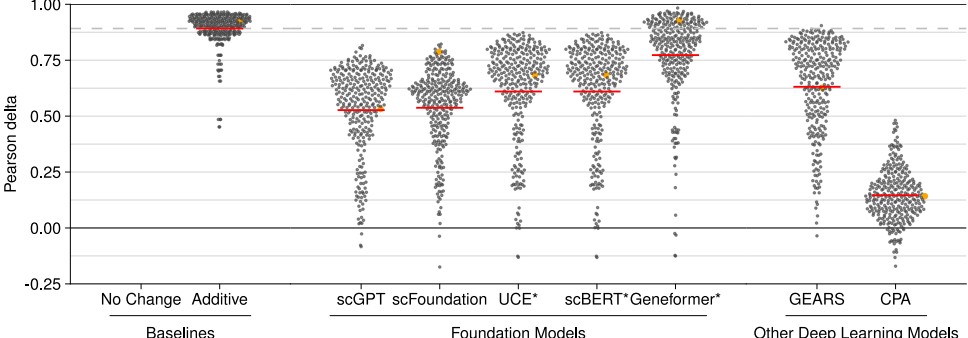

**(B) Prediction error stratified by the considered gene sets**

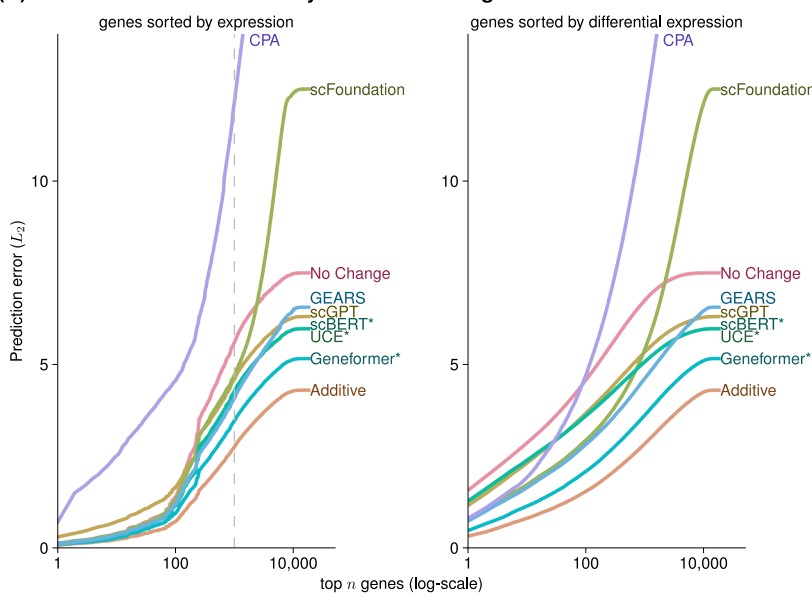

**Extended Data Fig. 2 | Alternative measures of the double perturbation prediction performance. (a)** The Pearson delta measure calculates the correlation of the prediction and observations after subtracting the expression in the control condition. The correlation for the *no change* predictions could not be calculated because they were all zero. The horizontal red lines show the mean per model, and the dashed line indicates the correlation of the best-performing model. **(b)** Prediction error as a function of *n*, the number of read-out genes. Left: genes ranked by expression in the control condition, right: by differential expression between observed value and expression in the control condition. Note that sorting by differential expression is only possible if access to the ground truth is available and can thus not be applied in real-world use cases. The dashed line at *n* = 1000 is the choice in Panel **a** and elsewhere in this work.

**(A) Quantile-Quantile plot of the difference from the additive expectation**

**(B) Empirical null decomposition**

**Extended Data Fig. 3 | Distribution of the observed difference from the additive model.** (**a**) Quantile-quantile plot comparing the distribution of the differences between observed expression values and the additive expectation against a standard normal distribution. The slope of the line is the standard deviation of the null model. (**b**) Histogram of the differences with a red curve overlayed that shows the null distribution fitted using *locfdr*. Values under the curve are grey, and the black bars show the observations that exceed what we would expect under the null model. The vertical bar shows the upper and lower thresholds for which the observations have a false discovery rate of less than 5% (that is, the grey fraction of the bars outside the vertical lines is 5%). The numbers at the top count the observations per group.

**(A) Precision-Recall Curve (PRC)**

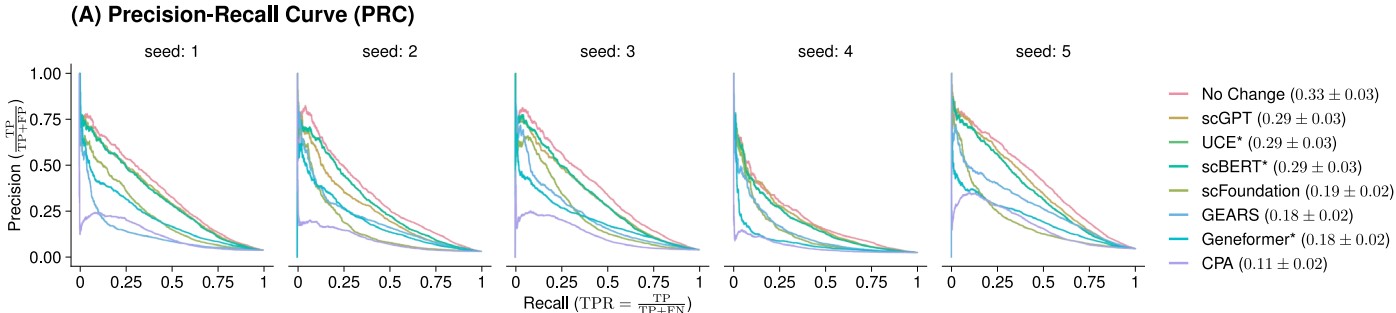

**(B) Receiver Operator Curve (ROC)**

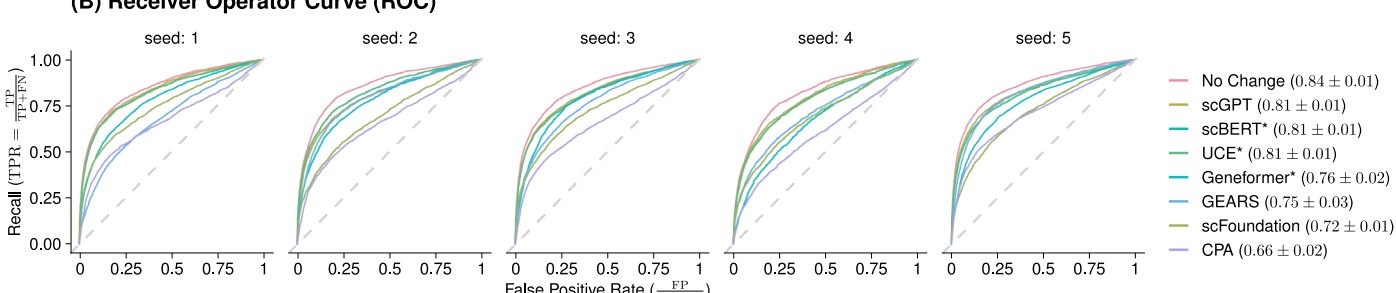

**Extended Data Fig. 4 | Alternative measures how well each model detects genetic interactions. (a)** Precision-recall and (**b**) receiver operator curve for all models distinguishing interactions from additive combinations. The numbers in parenthesis are the area under the curve (AUC) with the standard error across five test-training splits. *TP*: true positive, *FP*: false positive, *FN*: false negative, *TN*: true negative.

**(A) Reoccuring genes among top 100 interaction predictions**

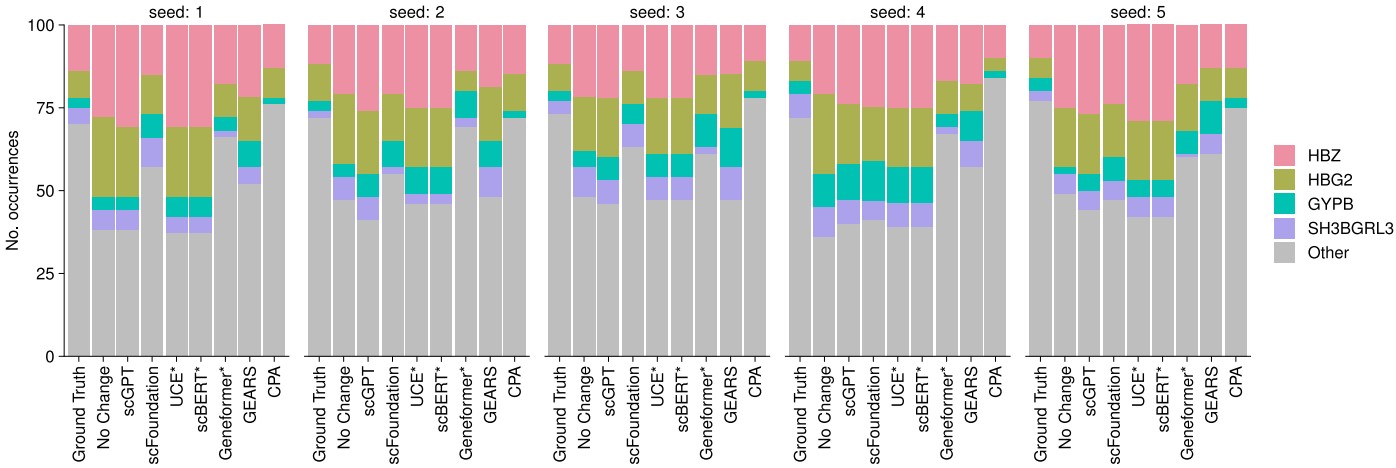

**(B) Reoccuring perturbations among top 100 interaction predictions**

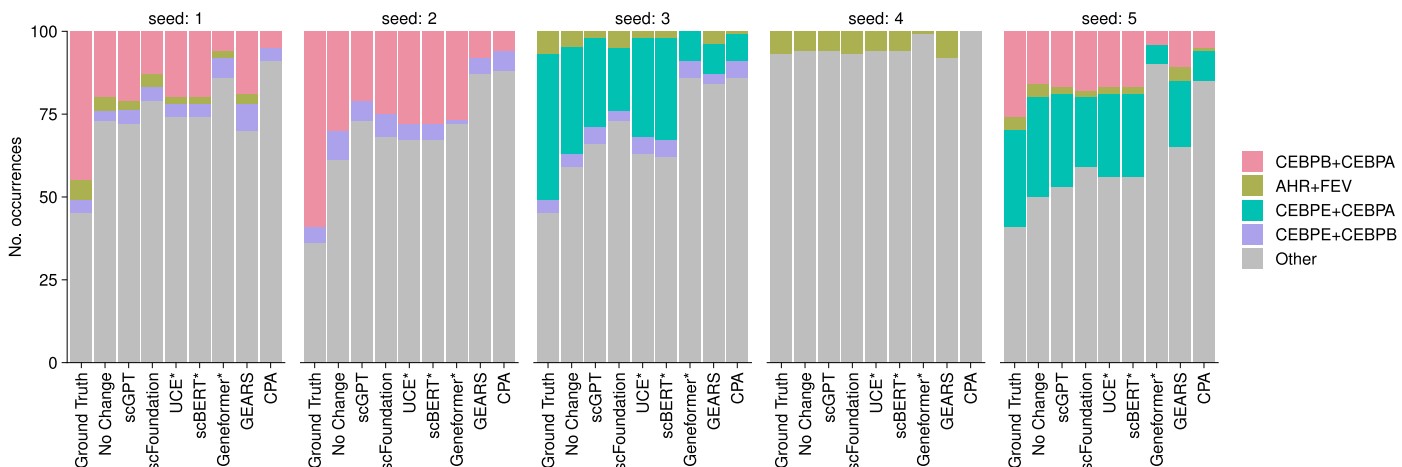

**Extended Data Fig. 5 | Reoccurrence of genes and perturbations for top predictions.** (**a**) Reoccurrence of genes and (**b**) perturbations among the 100 predictions that differed most from the additive expectation. The data is facetted by the test-training split. The ground truth column shows the genes and perturbations sorted by observed difference from the additive expectation. The highlighted genes and perturbations are the four most reoccurring ones.

## Analysis of the predicted and observed expression patterns for *HBG2* and *HBZ*

**Extended Data Fig. 6 | Comparison of predicted and observed expression for *HBG2* and *HBZ*.** Comparison of the predicted expression (black squares), the observed expression values (points colored by interaction type), and the range of values that are considered additive (grey boxes) for all test perturbations with seed = 1. The grey horizontal line shows the expression of *HBG2* and *HBZ* without perturbation.

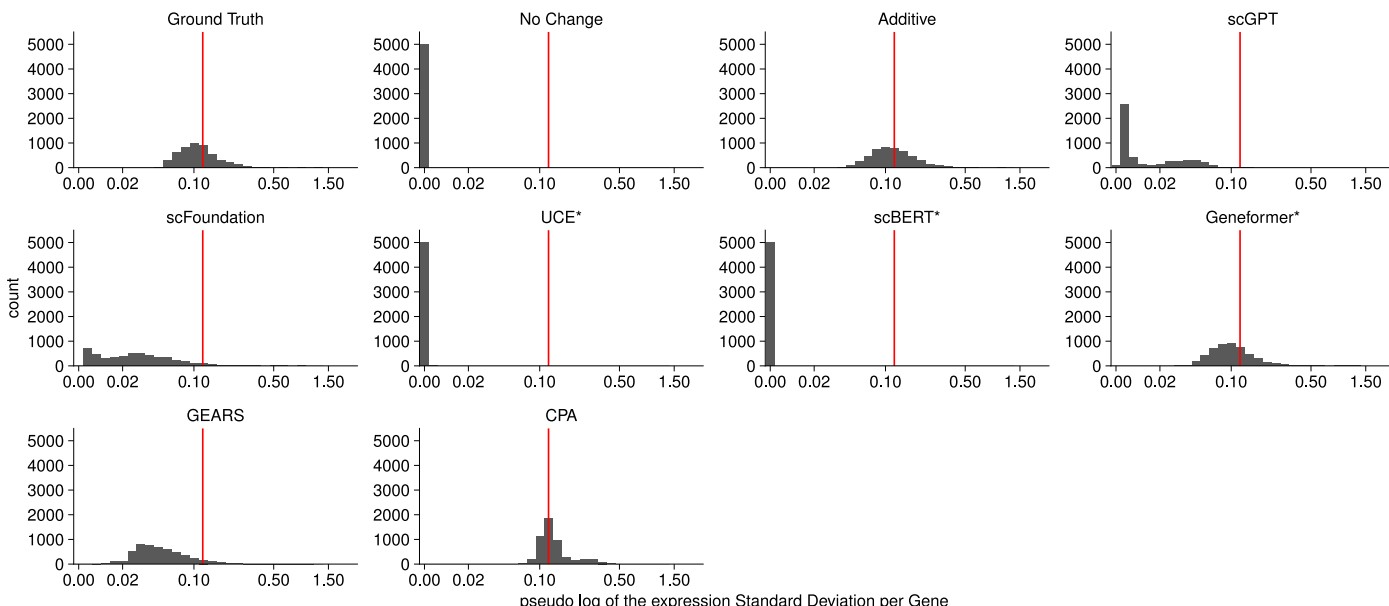

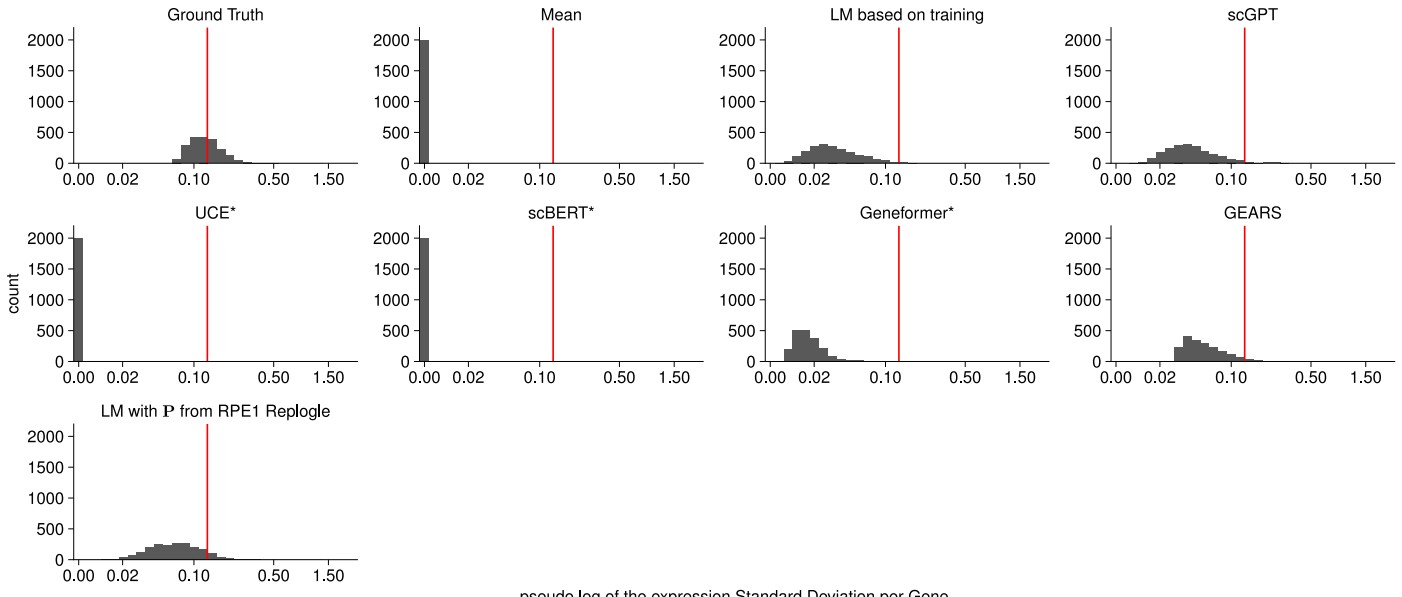

**Extended Data Fig. 7 | Variation of the predicted and observed expression values.** Histogram of the standard deviation per gene for the predicted and observed expression values across perturbations facetted by the model. The red vertical bar indicates the mean of the standard deviations for the ground truth for (**a**) the Norman dataset and (**b**) the Replogle K562 dataset. The data reflects the variation for the 1 000 most highly expressed genes and is aggregated across five test-training splits. *LM*: linear model.

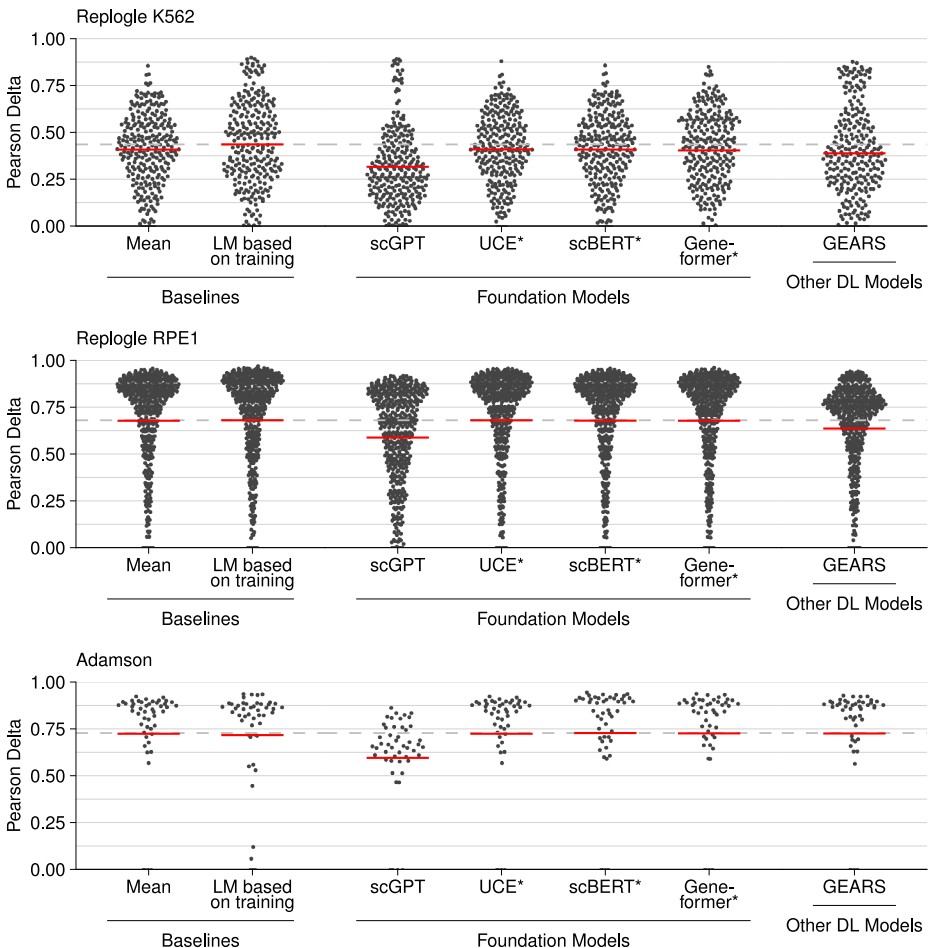

**(A) Single unseen perturbation prediction correlation**

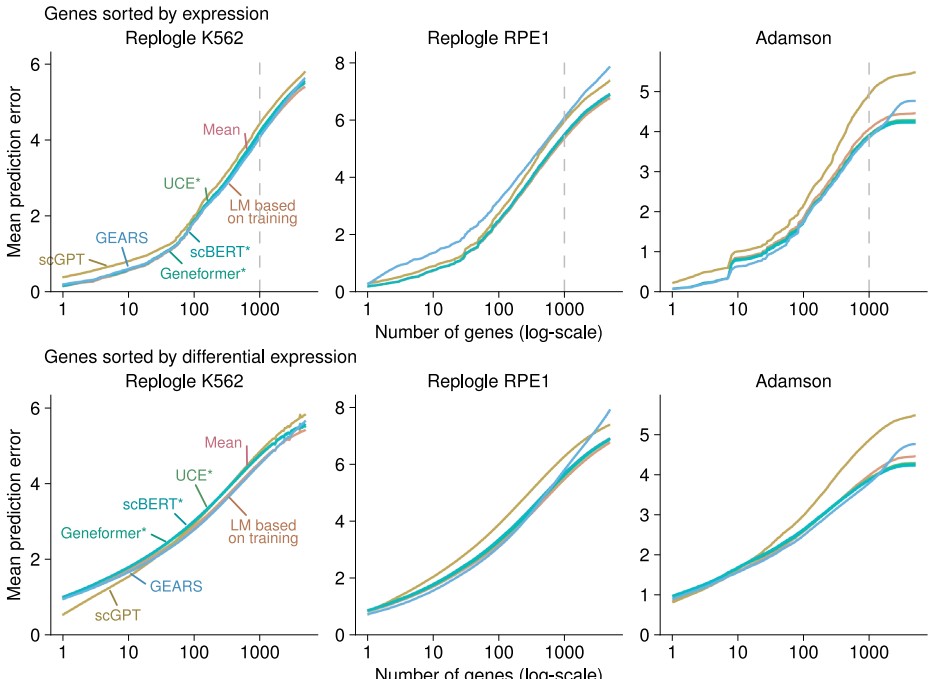

**(B) Prediction error stratified by the considered gene sets**

**Extended Data Fig. 8 | See next page for caption.**

**Extended Data Fig. 8 | Alternative measures of the single perturbation prediction performance.** (**a**) The Pearson delta measure calculates the correlation of the prediction and observations after subtracting the expression in the control condition. The horizontal red lines show the mean per model and the dashed line indicates the correlation of the best-performing model. (**b**) Prediction error as a function of $n$, the number of read-out genes.

Top: genes ranked by expression in the control condition. Bottom: by differential expression between observed value and expression in the control condition. Note that sorting by differential expression is only possible if access to the ground truth is available and can thus not be applied in real-world use cases. The dashed line at $n = 1000$ is the choice in Panel **a** and elsewhere in this work. *LM*: linear model, *DL*: deep learning.

**(A) Overall expression similarity of RPE1 and K562**

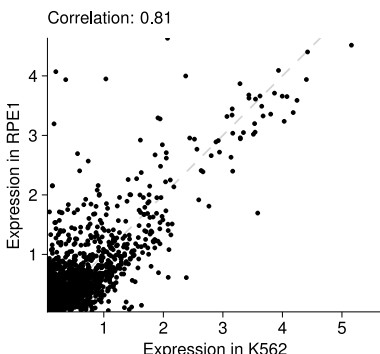

**(B) Read-out gene error depends its on diff. expression between K562 and RPE1**

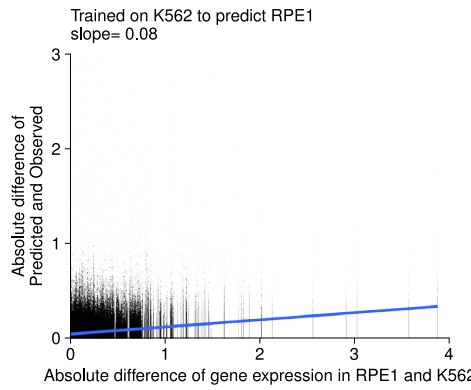

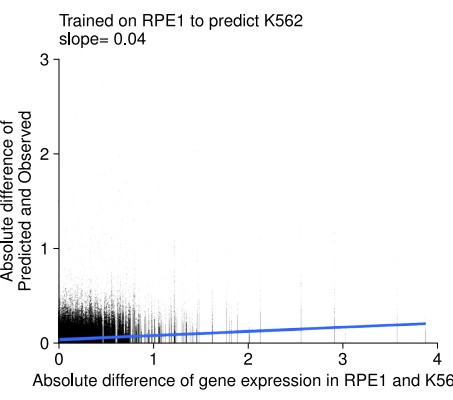

**(C) Perturbation correlation depends on the target gene diff. expression between K562 and RPE1**

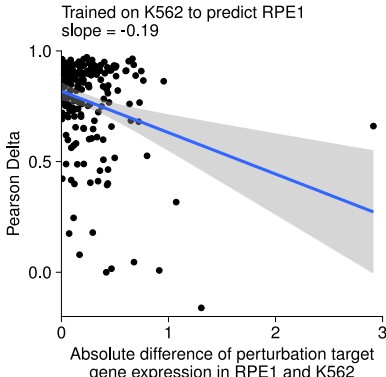

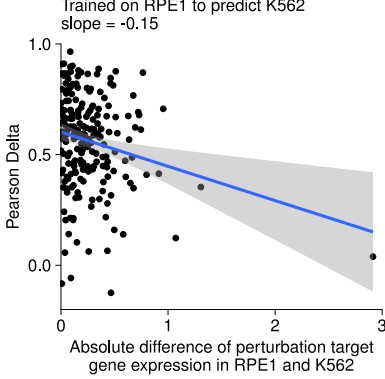

**Extended Data Fig. 9 | Analysis how differential expression between K562 and RPE1 effects prediction accuracy of transfer learning.** (**a**) Scatter plot of the mean gene expression for shared genes between RPE1 and K562 without perturbation. The dashed line indicates the diagonal. (**b**) Scatter plot of the absolute prediction error per read-out gene against the differential expression of that gene between RPE1 and K562. Each point is one read-out gene from one of the 122 double perturbations from five test-training splits. The blue line shows the linear fit with a slope indicated in the subtitle. (**c**) Scatter plot of the Pearson delta score per perturbation for the RPE1 dataset against the differential expression of the perturbation target gene between RPE1 and K562. The blue line shows the linear fit with a slope indicated in the subtitle, and the shaded area indicates the standard error of the fit.

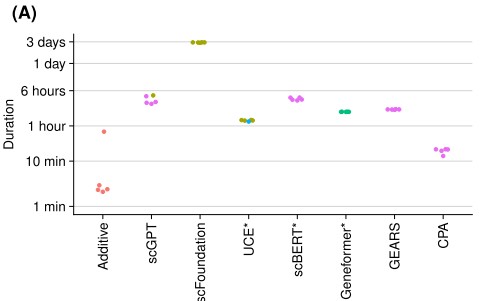

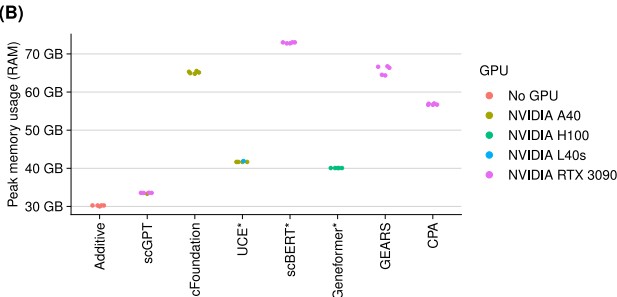

**Extended Data Fig. 10 | Computational resource requirements.** The resource usage was measured for the Norman dataset, which had 19 624 genes and 81 143 cells grouped into 225 conditions. Each point is one of the five test-training splits. (**a**) Elapsed time on a log scale to fine-tune and predict the double perturbations. (**b**) Peak memory usage for each model was measured using GNU time. The points are colored by the respective GPU model that was used.

# Reporting Summary

## Statistics

For all statistical analyses, confirm that the following items are present in the figure legend, table legend, main text, or Methods section.

| n/a | Confirmed | |
|---|---|---|
| ☐ | ☒ | The exact sample size (*n*) for each experimental group/condition, given as a discrete number and unit of measurement |
| ☒ | ☐ | A statement on whether measurements were taken from distinct samples or whether the same sample was measured repeatedly |
| ☒ | ☐ | The statistical test(s) used AND whether they are one- or two-sided<br>*Only common tests should be described solely by name; describe more complex techniques in the Methods section.* |
| ☒ | ☐ | A description of all covariates tested |
| ☒ | ☐ | A description of any assumptions or corrections, such as tests of normality and adjustment for multiple comparisons |
| ☐ | ☒ | A full description of the statistical parameters including central tendency (e.g. means) or other basic estimates (e.g. regression coefficient) AND variation (e.g. standard deviation) or associated estimates of uncertainty (e.g. confidence intervals) |
| ☒ | ☐ | For null hypothesis testing, the test statistic (e.g. *F*, *t*, *r*) with confidence intervals, effect sizes, degrees of freedom and *P* value noted<br>*Give P values as exact values whenever suitable.* |
| ☒ | ☐ | For Bayesian analysis, information on the choice of priors and Markov chain Monte Carlo settings |
| ☒ | ☐ | For hierarchical and complex designs, identification of the appropriate level for tests and full reporting of outcomes |
| ☐ | ☒ | Estimates of effect sizes (e.g. Cohen's *d*, Pearson's *r*), indicating how they were calculated |

*Our web collection on statistics for biologists contains articles on many of the points above.*

## Software and code

Policy information about availability of computer code

| | |
|---|---|
| Data collection | We did not use any software for data collection. |
| Data analysis | We used R version 4.4.1 and Python version 3.10.4 to run GEARS version 0.1.2, scGPT version 0.2.1,  scFoundation (build around GEARS version 0.0.2), CPA version 0.8.8, Geneformer version 0.1.0, scBERT from commit hash 262fd4b9 with model weights provided by the authors, and UCE at commit hash 8227a65c. We used igraph version 2.1.4 and locfdr 1.1-8. |

For manuscripts utilizing custom algorithms or software that are central to the research but not yet described in published literature, software must be made available to editors and reviewers. We strongly encourage code deposition in a community repository (e.g. GitHub). See the Nature Portfolio guidelines for submitting code & software for further information.

## Data

Policy information about availability of data

All manuscripts must include a data availability statement. This statement should provide the following information, where applicable:
- Accession codes, unique identifiers, or web links for publicly available datasets
- A description of any restrictions on data availability
- For clinical datasets or third party data, please ensure that the statement adheres to our policy

| Dataset | Availability (with link) |
|---|---|
| Norman | Downloaded from scFoundation via Figshare (https://figshare.com/articles/dataset/scFoundation_Large_Scale_Foundation_Model_on_Single- |

cell_Transcriptomics_-_processed_datasets/24049200?file=44477939)
Adamson          Downloaded via GEARS from Harvard dataverse (https://dataverse.harvard.edu/api/access/datafile/6154417)
Replogle K562   Downloaded via GEARS from Harvard dataverse (https://dataverse.harvard.edu/api/access/datafile/7458695)
Replogle RPE1   Downloaded via GEARS from Harvard dataverse (https://dataverse.harvard.edu/api/access/datafile/7458694)

## Research involving human participants, their data, or biological material

Policy information about studies with [human participants or human data](). See also policy information about [sex, gender (identity/presentation), and sexual orientation]() and [race, ethnicity and racism]().

| | |
|---|---|
| Reporting on sex and gender | N/A |
| Reporting on race, ethnicity, or other socially relevant groupings | N/A |
| Population characteristics | N/A |
| Recruitment | N/A |
| Ethics oversight | N/A |

Note that full information on the approval of the study protocol must also be provided in the manuscript.

# Field-specific reporting

Please select the one below that is the best fit for your research. If you are not sure, read the appropriate sections before making your selection.

☒ Life sciences          ☐ Behavioural & social sciences          ☐ Ecological, evolutionary & environmental sciences

For a reference copy of the document with all sections, see [nature.com/documents/nr-reporting-summary-flat.pdf](nature.com/documents/nr-reporting-summary-flat.pdf)

# Life sciences study design

All studies must disclose on these points even when the disclosure is negative.

| | |
|---|---|
| Sample size | We chose publicy available datasets whose experimental designs, incl. sample size, were appropriate as a test case for the proposed method / underlying scientific question. |
| Data exclusions | We did not exclude any data. |
| Replication | All material to replicate our results are available at https://github.com/const-ae/linear_perturbation_prediction-Paper/ and https://doi.org/10.5281/zenodo.14833202. However, we did not attempt to independently replicate our results. |
| Randomization | Not applicable. We present a complete combinatorial matrix of benchmarks of multiple computational methods each applied to multiple datasets and thus could observe each software in all conditions and did not need to randomize software to condition assignment. |
| Blinding | The analysts were not blinded while evaluating the benchmark. During the implementation of the benchmark, it is not possible to be blinded, as each method requires custom code. And while reporting the results blinding was not necessary as we directly provide the results for metrics that have previously already been used to measure the accuracy of perturbation effect prediction. To ensure that all methods were fairly evaluated, we approached the authors of each method and asked them to scrutinize our implementation. |

# Reporting for specific materials, systems and methods

We require information from authors about some types of materials, experimental systems and methods used in many studies. Here, indicate whether each material, system or method listed is relevant to your study. If you are not sure if a list item applies to your research, read the appropriate section before selecting a response.

## Materials & experimental systems

| n/a | Involved in the study |
|-----|----------------------|
| ☒ ☐ | Antibodies |
| ☒ ☐ | Eukaryotic cell lines |
| ☒ ☐ | Palaeontology and archaeology |
| ☒ ☐ | Animals and other organisms |
| ☒ ☐ | Clinical data |
| ☒ ☐ | Dual use research of concern |
| ☒ ☐ | Plants |

## Methods

| n/a | Involved in the study |
|-----|----------------------|
| ☒ ☐ | ChIP-seq |
| ☒ ☐ | Flow cytometry |
| ☒ ☐ | MRI-based neuroimaging |

## Plants

| Seed stocks | N/A |
|-------------|-----|
| Novel plant genotypes | N/A |
| Authentication | N/A |

