## [Transparent Peer Review file · Nature Methods]

Deep learning-based gene perturbation effect prediction does not yet outperform simple linear baselines

Corresponding Author: Dr Constantin Ahlmann-Eltze

This manuscript has been previously reviewed at another journal. This document only contains information relating to versions considered at Nature Methods.

Version 0:

Decision Letter:

2nd Dec 2024

Dear Dr Ahlmann-Eltze,

Your Brief Communication, "Deep learning-based predictions of gene perturbation effects do not yet outperform simple linear methods", has now been seen by 2 reviewers. As you will see from their comments below, although the reviewers find your work of potential interest, they have raised a number of concerns. We are interested in the possibility of publishing your paper in Nature Methods, but would like to consider your response to these concerns before we reach a final decision on publication.

We therefore invite you to revise your manuscript to address all these concerns. Please also make sure the fine-tuning procedure is well described and improve descriptions where necessary.

Link Redacted

We hope to receive your revised paper within 2 months. If you cannot send it within this time, please let us know. In this event, we will still be happy to reconsider your paper at a later date so long as nothing similar has been accepted for publication at Nature Methods or published elsewhere.

OPEN SCIENCE REQUIREMENTS

REPORTING SUMMARY AND EDITORIAL POLICY CHECKLISTS

DATA AVAILABILITY

All novel DNA and RNA sequencing data, protein sequences, genetic polymorphisms, linked genotype and phenotype data, gene expression data, macromolecular structures, and proteomics data must be deposited in a publicly accessible database, and accession codes and associated hyperlinks must be provided in the "Data Availability" section.

CODE AVAILABILITY

Please include a "Code Availability" subsection in the Online Methods which details how your custom code is made available. Only in rare cases (where code is not central to the main conclusions of the paper) is the statement "available upon request" allowed (and reasons should be specified).

MATERIALS AVAILABILITY

More details about our materials availability policy can be found at <https://www.nature.com/nature-portfolio/editorial->

ORCID

Sincerely,

Lin Tang, PhD
Senior Editor
Nature Methods

Reviewers' Comments:

Reviewer #1 (Remarks to the Author):

In this paper, Ahlmann-Eltze, Huber, and Anders present a critical benchmarking study comparing deep learning-based methods for predicting gene perturbation effects against simple linear approaches. They specifically evaluate two state-of-the-art foundation models (scGPT and scFoundation) and a graph-based deep learning framework (GEARS) in two key scenarios: predicting combinatorial perturbation effects and predicting outcomes of unseen perturbations. Through careful analysis, they demonstrate that simple linear models perform comparably or better than these more complex approaches, highlighting important considerations for the field. Before publication it will be important to consider these points:

1. It would be valuable to explore whether there are specific interactions that are not captured by the linear model but are captured by foundational models. The authors should dig deeper into these interactions to see if the models have specific value in uncovering interactions related to biological processes or gene sets, rather than just analyzing performance metrics over gene expression values.
2. To ensure fairness in the comparison, the authors should consider contacting the developers of these foundational models to verify that the parameters and implementation choices are appropriate for this benchmark. Given that the code is provided on GitHub (which is excellent!), consulting with original authors would ensure the models are being run optimally for this comparison.
3. The manuscript should better explain why the scFoundation model was not included in the single perturbation prediction experiments. The current explanation about requiring specially preprocessed input data needs elaboration - is this due to missing code from the original authors, computational constraints, or other practical limitations? This practical information would be valuable for users trying to implement these methods.
4. For the transfer learning approach in single perturbation prediction (training on one cell type and predicting another), it would be valuable to analyze whether prediction is more challenging for genes that are already differentially expressed between cell lines, especially for master regulators and transcription factors whose perturbation should affect many other genes. This kind of analysis could reveal important biological insights about the models' limitations.
5. The claim that deep learning models don't provide any performance benefits over simple linear prediction should be reconsidered, as the authors acknowledge that using embeddings from pre-trained models can boost performance in some tasks. This suggests there may be value in the transformer-based architecture and these embeddings might capture properties that simple linear models (like PCA-based approaches) don't capture. While the originally proposed fine-tuning procedures might be underdeveloped, the learned embeddings show promise.

Overall, the paper is well-written and will be very impactful for the field, potentially helping to temper overselling statements in previous publications. The inclusion of source code for the benchmark is particularly valuable. With the suggested additions and clarifications, it will provide even more comprehensive insights for the community.

Reviewer #2 (Remarks to the Author):

The manuscript evaluates the predictive performance of deep learning-based approaches for gene perturbation outcomes against simpler linear models. It highlights the current limitations of deep learning in outperforming simpler models for tasks

such as combinatorial and unseen perturbation prediction. While the manuscript addresses an important and timely topic, there are several areas that require further clarification, additional experiments, and enhanced discussion. Below are detailed comments and suggestions.

Major Comments

1. To strengthen the manuscript's conclusions, it is recommended to include additional deep learning models in the comparison. For example, integrating state-of-the-art models such as GeneFormer (Theodoris et al., 2023), scBERT (Yang et al., 2022), UCE (Rosen et al., 2023), xTrimoGene (Gong et al., 2023), scVI (Lopez et al., 2023), and CINEMA-OT (Dong et al., 2023) could provide a broader perspective on the current capabilities of deep learning approaches in gene perturbation prediction. Including these models would enhance the robustness of the findings and demonstrate whether the limitations identified are specific to the tested models or reflect broader trends in deep learning for this task.

2. The manuscript relies on datasets from different studies but provides limited descriptions of their characteristics. For example, the magnitude of changes in gene expression after perturbations may bias the results. A detailed summary table describing key features (e.g., sample size, cell type, gene set coverage, changes in gene expression and noise levels) would improve clarity and help contextualize performance comparisons. Additionally, discuss how variability in dataset properties—such as experimental noise, environmental conditions, or the diversity of perturbations—impacts model outcomes and generalizability. This discussion would enhance the manuscript's scientific rigor.

3. The findings indicate that linear models perform on par with or better than deep learning approaches in most benchmarks. However, the potential biological or methodological reasons for this result are not fully explored. For instance, in Figure 1D, none of the tested methods excel at predicting synergistic interactions, and the "no change" model outperforms deep learning-based approaches when identifying genes with synergistic perturbation effects. Could this be due to overfitting in the deep models, lack of sufficient training data, or inherent simplicity in the perturbation patterns in the datasets? Or do deep learning models tend to default to predicting no change relative to the control condition? Including ablation studies on model architectures, training data size, and a quantitative analysis of prediction outcomes would provide deeper insights.

4. Could authors clarify why specific metrics, such as Pearson delta, were chosen for benchmarking tasks. Additionally, consider including other metrics such as R^2 , similarity of log fold change, RMSE or scPerturb to provide a more comprehensive assessment of model performance.

5. Supplementary Figure S6 highlights the computational inefficiencies of deep learning models. Specifically, information such as the GPU version used, minimum hardware requirements, runtime for different benchmarks, and peak memory usage should be provided.

6. The manuscript provides a valuable critique of the limitations of current deep learning methods. Expanding on how these limitations might be addressed in future work (e.g., better pre-training, incorporating biological priors) would provide more actionable insights for the field and guide future research.

Minor Comments

7. Proofreading is recommended to address minor typographical errors (e.g., "additve" in Fig. 1 caption).

8. Ensure consistency in citation formatting throughout the manuscript. For example, reference [11] uses brackets, while others use superscripts.

This manuscript provides valuable insights into the challenges of applying deep learning to biological perturbation prediction. It raises important questions about the current utility of deep learning and lays the groundwork for future model improvements. Addressing the above comments will enhance the manuscript's clarity, rigor, and contribution to the field.

Version 1:

Decision Letter:

Our ref: NMETH-BC58232A

31st Mar 2025

Dear Dr. Ahlmann-Eltze,

Thank you for submitting your revised manuscript "Deep learning-based predictions of gene perturbation effects do not yet outperform simple linear baselines" (NMETH-BC58232A). It has now been seen by the original referees and their comments are below. The reviewers find that the paper has improved in revision, and therefore we'll be happy in principle to publish it in Nature Methods, pending minor revisions to comply with our editorial and formatting guidelines.

We are now performing detailed checks on your paper and will send you a checklist detailing our editorial and formatting

requirements within two weeks or so. Please do not upload the final materials and make any revisions until you receive this additional information from us.

TRANSPARENT PEER REVIEW

ORCID

Sincerely,

Lin Tang, PhD
Senior Editor
Nature Methods

Reviewer #1 (Remarks to the Author):

The authors have addressed all my concerns. I look forward to seeing the manuscript published.

Reviewer #2 (Remarks to the Author):

Thank you for addressing my concerns. This work will be a valuable contribution to the field, providing a reliable benchmark for predicting the effects of genetic perturbations.

Version 2:

Decision Letter:

24th Jun 2025

Dear Dr Ahlmann-Eltze,

Thank you very much for sending us the updated files of your Brief Communication "Deep learning-based gene perturbation effect prediction does not yet outperform simple linear baselines". I am pleased to inform you that this paper has now been accepted for publication in Nature Methods. The received and accepted dates will be 11th Oct 2024 and 24th Jun 2025. This note is intended to let you know what to expect from us over the next month or so, and to let you know where to address any further questions.

Over the next few weeks, your paper will be copyedited to ensure that it conforms to Nature Methods style. Once your paper is typeset, you will receive an email with a link to choose the appropriate publishing options for your paper and our Author Services team will be in touch regarding any additional information that may be required.

Once proofs are generated, they will be sent to you electronically and you will be asked to send a corrected version within 48 hours. It is extremely important that you let us know now whether you will be difficult to contact over the next month. If this is the

case, we ask that you send us the contact information (email, phone and fax) of someone who will be able to check the proofs and deal with any last-minute problems.

If, when you receive your proof, you cannot meet the deadline, please inform us at rjsproduction@springernature.com immediately.

To assist our authors in disseminating their research to the broader community, our SharedIt initiative provides you with a unique shareable link that will allow anyone (with or without a subscription) to read the published article. Recipients of the link with a subscription will also be able to download and print the PDF. As soon as your article is published, you will receive an automated email with your shareable link.

Please note that you and your coauthors may order reprints and single copies of the issue containing your article through Springer Nature Limited's reprint website, which is located at <http://www.nature.com/reprints/author-reprints.html>. If there are any questions about reprints please send an email to author-reprints@nature.com and someone will assist you.

Please feel free to contact me if you have questions about any of these points. Thank you very much for publishing your paper at Nature Methods!

Best regards,

Lin Tang, PhD
Senior Editor
Nature Methods

** Visit the Springer Nature Editorial and Publishing website at [http://www.springernature.com/editorial-and-publishing-jobs?](http://www.springernature.com/editorial-and-publishing-jobs?utm_source=ejP_NMeth_email&utm_medium=ejP_NMeth_email&utm_campaign=ejp_Nmeth) www.springernature.com/editorial-and-publishing-jobs for more information about our career opportunities. If you have any questions please click [here](mailto:editorial.publishing.jobs@springernature.com).

Point-by-point response

Deep learning-based predictions of gene perturbation effects do not yet outperform simple linear baselines

Reviewer 1

In this paper, Ahlmann-Eltze, Huber, and Anders present a critical benchmarking study comparing deep learning-based methods for predicting gene perturbation effects against simple linear approaches. They specifically evaluate two state-of-the-art foundation models (scGPT and scFoundation) and a graph-based deep learning framework (GEARS) in two key scenarios: predicting combinatorial perturbation effects and predicting outcomes of unseen perturbations. Through careful analysis, they demonstrate that simple linear models perform comparably or better than these more complex approaches, highlighting important considerations for the field.

We thank the reviewer for the time and effort they spent on our manuscript and for their positive feedback.

Before publication it will be important to consider these points:

1. It would be valuable to explore whether there are specific interactions that are not captured by the linear model but are captured by foundational models. The authors should dig deeper into these interactions to see if the models have specific value in uncovering interactions related to biological processes or gene sets, rather than just analyzing performance metrics over gene expression values.

We do not consider our baselines to be serious competitors for predicting perturbation effects. The additive model can, by definition, not find any interactions. And the *no change* baseline predictions are independent of the training data and can thus not learn anything interesting about the data. We have updated the text to clarify this.

We have added a section where we explore the differences in the type of interactions each model discovers (Fig. 1F). We find that all models are much better at identifying the deviation from the additive model if the effect is buffering. We further quantified how often the models predict synergistic effects among the 500 predictions furthest from the additive expectation, as this is something that neither one of our baselines could do. However, we

find (1) that the deep learning models rarely predict synergistic effects and (2) that it is even rarer that those predictions are correct (Fig. 1F bottom panel). We also found no obvious biological pattern among the correct predictions of synergistic interaction.

Figure 1: (A) Overlap of top 500 interaction predictions per model. The blue bars show the interactions predicted by at least three deep learning tools but not the *no change* baseline. (B) The most recurring genes and perturbations among the blue sets

We further checked for interesting biological patterns among the genes identified by a subset of the deep learning methods but not our *no change* baseline (see Figure 1). However, we found a similar set of genes and perturbations to the ones we identified across all methods in Suppl. Fig. S5.

2. To ensure fairness in the comparison, the authors should consider contacting the developers of these foundational models to verify that the parameters and implementation choices are appropriate for this benchmark. Given that the code is provided on GitHub (which is excellent!), consulting with original authors would ensure the models are being run optimally for this comparison.

We thank the reviewer for the suggestion. We contacted the first and last authors of all benchmarked methods and received replies from the authors of Geneformer, scFoundation, and CPA.

- The authors of Geneformer emphasized that the model is not designed or tested for predicting the expression changes after perturbing individual genes *in silico*. They said they expected the model to simply return the original expression values. They also emphasized that during the training of Geneformer cancer cell lines were excluded because of concern over the large number of mutations, that for many Perturb-seq datasets the target gene shows little expression change, and that they used ChIP-seq data to get a clean mapping of the direct targets of a perturbed transcription factor.
- The authors of scFoundation did not find any problem with our code and acknowledged that the current models are still far from satisfactory. They also pointed to their benchmark preprint (<https://doi.org/10.1101/2024.12.20.629581>) that shows mixed

results when comparing scFoundation and other foundation models against simple baselines.

- The authors of CPA perceived a problem in our code and submitted a pull request to fix the issue (https://github.com/const-ae/linear_perturbation_prediction_Paper/pull/3). However, when we ran the updated code, the performance of CPA was worse than with our original version of the code. In the paper, we report the better-for-CPA (initial) result, and explain the issue in the Methods section.

3. The manuscript should better explain why the scFoundation model was not included in the single perturbation prediction experiments. The current explanation about requiring specially preprocessed input data needs elaboration - is this due to missing code from the original authors, computational constraints, or other practical limitations? This practical information would be valuable for users trying to implement these methods.

We have added a sentence to explain the problem. scFoundation requires the genes of the input data to exactly match the set of genes from their pre-training dataset. For the Replogle and Adamson data, 14,000 of the required genes were missing. We could have “hacked” this, e.g., by setting the expression to zero for all of these genes, but then it would be difficult to conclude if any performance deficits were due to inappropriate input data or inherent limitations of the model, so we decided to circumvent this rabbit hole and not use scFoundation for the analyses reported in Figure 2.

4. For the transfer learning approach in single perturbation prediction (training on one cell type and predicting another), it would be valuable to analyze whether prediction is more challenging for genes that are already differentially expressed between cell lines, especially for master regulators and transcription factors whose perturbation should affect many other genes. This kind of analysis could reveal important biological insights about the models’ limitations.

We thank the reviewer for the suggestion. We confirmed their hypothesis that the differential expression between K562 and RPE1 was predictive of the prediction error per gene. We also found a strong negative association between the differential expression of the perturbation target gene and the Pearson Delta for that perturbation. We included this analysis in Suppl. Fig. S9.

5. The claim that deep learning models don’t provide any performance benefits over simple linear prediction should be reconsidered, as the authors acknowledge that using embeddings from pre-trained models can boost performance in some tasks. This suggests there may be value in the transformer-based architecture, and these embeddings might capture properties that simple linear models (like PCA-based approaches) don’t capture. While the originally proposed fine-tuning procedures might be underdeveloped, the learned embeddings show promise.

We thank the reviewer for this suggestion. It appears to have been based on an—understandable—misinterpretation of Figure 2C in the original manuscript, which, we admit, was confusing. We apologize for this. We have now updated Figure 2C, and instead of individual plots for the different methods, it now shows a single forest plot of the performance relative to the baseline (“mean”). From this, one can see that the transformer-derived embeddings do not consistently outperform the linear model learned from the training data.

Overall, the paper is well-written and will be very impactful for the field, potentially helping to temper overselling statements in previous publications. The inclusion of source code for the benchmark is particularly valuable. With the suggested additions and clarifications, it will provide even more comprehensive insights for the community.

Thank you!

Reviewer 2

The manuscript evaluates the predictive performance of deep learning-based approaches for gene perturbation outcomes against simpler linear models. It highlights the current limitations of deep learning in outperforming simpler models for tasks such as combinatorial and unseen perturbation prediction. While the manuscript addresses an important and timely topic, there are several areas that require further clarification, additional experiments, and enhanced discussion. Below are detailed comments and suggestions.

We thank the reviewer for their time and effort they spent on our manuscript, and for their positive feedback.

Major Comments

1. To strengthen the manuscript’s conclusions, it is recommended to include additional deep learning models in the comparison. For example, integrating state-of-the-art models such as GeneFormer (Theodoris et al., 2023), scBERT (Yang et al., 2022), UCE (Rosen et al., 2023), xTrimGene (Gong et al., 2023), scVI (Lopez et al., 2023), and CINEMA-OT (Dong et al., 2023) could provide a broader perspective on the current capabilities of deep learning approaches in gene perturbation prediction. Including these models would enhance the robustness of the findings and demonstrate whether the limitations identified are specific to the tested models or reflect broader trends in deep learning for this task.

We thank the reviewer for these suggestions. We added Geneformer, scBERT and UCE to our benchmark.

We could not include scVI because predicting the effects of unseen double or single perturbations is not a documented functionality of this software, and it is not immediately apparent how it could be used for that purpose. Instead, we included CPA, which is based on the same autoencoder idea as scVI, directly supports predicting expression changes with unseen double perturbations, and has been used as a baseline by GEARS.

We also did not include xTrimoGene, as it is the predecessor of scFoundation, and we could not locate public code that would enable us to run it.

We decided against including CINEMA-OT, as it is not based on deep learning, and as such, not in the scope of this paper.

2. The manuscript relies on datasets from different studies but provides limited descriptions of their characteristics. For example, the magnitude of changes in gene expression after perturbations may bias the results. A detailed summary table describing key features (e.g., sample size, cell type, gene set coverage, changes in gene expression and noise levels) would improve clarity and help contextualize performance comparisons. Additionally, discuss how variability in dataset properties—such as experimental noise, environmental conditions, or the diversity of perturbations—impacts model outcomes and generalizability. This discussion would enhance the manuscript’s scientific rigor.

We thank the reviewer for the suggestion and included a detailed description of each dataset in Suppl. Fig. S1. It contains an overview of the size of the datasets, a 2D representation of the perturbation similarities, and a plot that shows the change in expression of the perturbation target genes.

We also amended the Discussion to mention that all datasets are from cancer cell lines, which have a higher mutational burden than healthy tissue, and that this could thus influence the performance of the foundation models.

3. The findings indicate that linear models perform on par with or better than deep learning approaches in most benchmarks. However, the potential biological or methodological reasons for this result are not fully explored. For instance, in Figure 1D, none of the tested methods excel at predicting synergistic interactions, and the “no change” model outperforms deep learning-based approaches when identifying genes with synergistic perturbation effects. Could this be due to overfitting in the deep models, lack of sufficient training data, or inherent simplicity in the perturbation patterns in the datasets? Or do deep learning models tend to default to predicting no change relative to the control condition? Including ablation studies on model architectures, training data size, and a quantitative analysis of prediction outcomes would provide deeper insights.

Instead of saying that the “linear (or no change) models perform on par with or better than deep learning approaches,” we prefer to say that “the tested deep learning approaches do not outperform these simple baseline models”. We have revised the manuscript to make this clearer. This means that we do not consider the linear (or no change) models to be serious

competitors. Rather, they are a baseline against which to compare the more sophisticated approaches.

We do see some evidence that the poor performance of scGPT, UCE and scBERT could, in part, be because they—as the reviewer notes—simply tend to default to predicting no change. In line with this, the predictions of scFoundation and GEARS vary significantly less than the observed expression values across perturbations. We document this in the new Suppl. Fig. S7.

In addition, we find that the fine-tuned decoders of scGPT, scFoundation, and GEARS perform worse than a simple linear model that uses the gene for scGPT and scFoundation or perturbation embeddings for GEARS (Figure 2C). We further demonstrate that training on another perturbation dataset consistently improves the prediction performance. This leads us to hypothesize that more pertinent training data would benefit the foundation models for this task.

A meaningful series of ablation studies of the foundation models is beyond the scope of this paper (see the table with pretraining requirements). We believe that justifying the architecture of these models or their possible improvement is primarily a task for the developers of these models, or the subject of substantial and lengthy follow-up research projects by others. We hope that our current paper can provide a motivation and help justify the resources for such work.

Pretraining requirements for the foundation models

Model	Number GPUs	Duration
scGPT	16 A100	6 days
scFoundation	64 A100	14 days
scBert	32 V100	7-14 days
Geneformer	12 V100	3 days
UCE	24 A100	40 days

4. Could authors clarify why specific metrics, such as Pearson delta, were chosen for benchmarking tasks. Additionally, consider including other metrics such as R^2 , similarity of log fold change, RMSE or scPerturb to provide a more comprehensive assessment of model performance.

We added a sentence to the Method section explaining the pros and cons of the L2 distance and Pearson delta metrics. The R^2 is equal to the squared correlation coefficient and thus would not offer additional insights into the performance. We have also added a sentence to clarify that the RMSE is just another name for the L2 distance. The E-distance, which was proposed in the scPerturb publication (Peidli et al. 2024), is a summary measure for the magnitude of change in single cell gene expression data between two conditions (e.g., control and perturbation). It is not designed to compare predicted and observed expression values.

5. Supplementary Figure S6 highlights the computational inefficiencies of deep learning models. Specifically, information such as the GPU version used, minimum hardware requirements, runtime for different benchmarks, and peak memory usage should be provided.

We updated Supplemental Data Figure S10, which shows the runtime and peak memory usage, to also reflect the GPU that was used to fine-tune and run each method.

6. The manuscript provides a valuable critique of the limitations of current deep learning methods. Expanding on how these limitations might be addressed in future work (e.g., better pre-training, incorporating biological priors) would provide more actionable insights for the field and guide future research.

We extended the Discussion with our hypothesis that one problem might be the observational nature of the data used for pre-training.

Minor Comments

7. Proofreading is recommended to address minor typographical errors (e.g., “ad-ditve” in Fig. 1 caption).

Fixed

8. Ensure consistency in citation formatting throughout the manuscript. For example, reference [11] uses brackets, while others use superscripts.

Fixed.

This manuscript provides valuable insights into the challenges of applying deep learning to biological perturbation prediction. It raises important questions about the current utility of deep learning and lays the groundwork for future model improvements. Addressing the above comments will enhance the manuscript’s clarity, rigor, and contribution to the field.

Thank you!